# Prospects and Challenges of the Study of Anti-Glycan Antibodies and Microbiota for the Monitoring of Gastrointestinal Cancer

**DOI:** 10.3390/ijms222111608

**Published:** 2021-10-27

**Authors:** Eugeniy P. Smorodin

**Affiliations:** Department of Virology and Immunology, National Institute for Health Development, 11619 Tallinn, Estonia; jevgeni.smorodin@outlook.com

**Keywords:** glycoconjugates, antibodies, cancer, survival, prognosis, tumour microbiota

## Abstract

Over the past decades, a large amount of data has been accumulated in various subfields of glycobiology. However, much clinically relevant data and many tools are still not widely used in medicine. Synthetic glycoconjugates with the known structure of glycans are an accurate tool for the study of glycan-binding proteins. We used polyacrylamide glycoconjugates (PGs) including PGs with tumour-associated glycans (TAGs) in immunoassays to assess the prognostic potential of the serum level of anti-glycan antibodies (AG Abs) in gastrointestinal cancer patients and found an association of AG Abs with survival. The specificity of affinity-isolated AG Abs was investigated using synthetic and natural glycoconjugates. AG Abs showed mainly a low specificity to tumour-associated and tumour-derived mucins; therefore, the protective role of the examined circulating AG Abs against cancer remains a challenge. In this review, our findings are analysed and discussed in the context of the contribution of bacteria to the AG Abs stimulus and cancer progression. Examples of the influence of pathogenic bacteria colonising tumours on cancer progression and patient survival through mechanisms of interaction with tumours and dysregulated immune response are considered. The possibilities and problems of the integrative study of AG Abs and the microbiome using high-performance technologies are discussed.

## 1. Introduction

Simple and complex carbohydrates (glycans, glycoconjugates) play a vital role in organisms. Glycans are targets for the recognition of self and foreign antigens by the immune system, and microorganisms use host glycans for their adhesion and further colonisation. The biological role and numerous functions of glycans have been well presented by Ajit Varki [1]. Abnormal structures of glycans are expressed in cancer that are promising biomarkers and targets for therapy. However, changes in the glycosylation pattern make the tumour cells evade immunosurveillance [2]. In this respect, the role of anti-glycan antibodies (AG Abs) in the mechanisms of anticancer defence remains unclear.

The repertoire of the circulating AG Abs in healthy individuals is relatively unique and stable, which makes it possible to monitor changes of the protective Abs in diseases [3]. The profile of AG Abs depends on contacts with environmental factors and past illnesses and represents Abs against foreign antigens and auto-Abs, including Abs that bind to tumour-associated glycans (TAGs) [4]. The circulating auto-Abs to TAGs might protect against cancer; however, their protective role and association with cancer progression remain poorly understood. We undertook the follow-up study of gastric and colorectal cancer (CRC) patients and used an enzyme-linked immunosorbent assay (ELISA) with polyacrylamide glycoconjugates (PGs) to monitor the levels of IgG Abs reactive to tumour-associated Thomsen–Friedenreich (TF) and its precursor (Tn), and other glycans to assess the association of the AG Abs level with survival and clinical parameters. The increased attention to these TAGs and respective Abs is due to their expression in malignant tumours; the relation to differentiation, invasiveness, and metastasis; as well as the potential for diagnostics, prognosis, and immunotherapy [2,5,6,7,8]. The expression of the so-called Thomsen–Friedenreich (TF) on red blood cells after treatment with bacterial neuraminidases was described by Thomsen and specified by Friedenreich [9]. In basic research by the G. Springer group, TF (in their abbreviation “T”) and Tn have been described as immunoreactive pancarcinoma antigens that are often expressed in tumours but are usually hidden and inaccessible to the immune system in normal tissues [10,11]. Furthermore, we studied the specificity of AG Abs isolated from the serum of long-term survivors. This paper analyses the research results and discusses the possibilities and challenges of an integrative approach to the study of AG Abs and pathogenic bacteria colonising tumours and their involvement in the progression of cancer and dysregulation of the immune response in the tumour microenvironment.

## 2. Association of the Level of AG Abs with Survival

The discovery of TAGs markers, such as CA 19-9 and CA 72-4, which are overexpressed in cancer cells and tumour samples and appear in the bloodstream, contributed to their widespread use in clinical oncology for diagnostic and prognostic purposes [12]. To the best of our knowledge, over the past decades, many works on diagnostic and predictive studies of TAGs have been published [13,14,15]. Glycan-specific antibodies are also represented in the literature as potential biomarkers for cancer diagnosis [16,17]. However, the relationship between AG Abs and prognosis of cancer patients remains poorly understood. A Search on PubMed, Web of Science, Scopus, and Crossref found only one noteworthy clinical investigation, i.e., the observation of patients with cervical cancer over a long period. The authors note that the presence of IgG AG Abs in sera was associated with better survival and treatment modalities [18]. We studied the level of IgG Abs that are reactive to TAGs, namely anti-TFα, -Tn, and cancer-irrelevant xenoreactive anti-αGal, as well as anti-GalNAcβ and -PF_di_ (structures shown in Table 1), in the sera of patients with gastric, colorectal, and breast cancer [8,19,20,21,22]. The preoperative level of Abs, its postoperative changes during a long-term follow-up, relation to survival, and clinical parameters were investigated [20,21]. A significantly better survival rate was observed in gastrointestinal cancer patients with an increased level of anti-TF, -Tn, and -GalNAcβ Abs, whereas a significantly worse survival rate was observed in groups with an increased level of anti-αGal Abs (Figure 1). The increased anti-PF_di_ IgG level was associated with the stages of gastric cancer, but no significant difference in survival rate was found [21,23]. 

The benefit from a higher level of anti-TF Abs in relation to survival might be explained through possible involvement of Abs in the inhibition of metastasis-promoting interaction of TF-MUC1 with galectins for circulating tumour cells. TF-MUC1 contributes to galectin-mediated adhesion of cancer cells to blood vascular endothelium. The increased level of circulating galectin-2 is correlated with a significantly increased mortality in patients with colorectal cancer (CRC). However, this association was not observed in patients with a high level of auto-Abs against TF-glycosylated MUC1 [24]. The role of galectins in cancer progression, including prognostic assessment in meta-analyses, has been published [25]. In addition to TF, another galectin 3 ligand, namely, the Lac-di-NAc glycan (GalNAcβ1-4GlcNAc), is also considered a TAG [26,27,28]. The role of interaction of Lac-di-NAc with galectin-3 in the context of circulating Abs in cancer progression remains unexplored.

The level of AG IgGs did not undergo major changes in most patients, but we observed more than a two-fold increase or decrease of Abs in dynamics (Table 1) [20,23,29,30]. A high level of anti-TF IgG was observed in 6% of long-term survivors, and among them, the survival rate exceeding 15 years was noted in stage III of gastrointestinal cancer. The increase of the anti-TF IgG level was noted after surgery, blood transfusion, influenza, and prolonged adjuvant chemotherapy. In the latter case, up to a 12-fold increase from the initial low background level was noted. Similarly to anti-TF IgG, the levels of anti-Tn and αGal IgG were increased after surgery or chemotherapy in some patients and remained elevated or returned to baseline values. This may be explained by the paradigm of gut microbiota involvement in AG Abs formation, which is discussed in Section 5, since both radical surgery and chemotherapy affect the gut microbiota [31].

The low preoperative level of anti-TF IgG remained low after surgery was performed on patients with metastatic cancer and prevailed in patients with poorly differentiated G3 tumours. On the contrary, an increased anti-αGal IgG level was related to advanced cancer [20,29,32]. According to a study by Springer et al., surgery leads to an increase of anti-TF antibody titres in breast cancer patients [33]. The level of anti-TF IgG was correlated with the number of lymphocytes in the blood and was inversely correlated with the ratio of neutrophils/lymphocytes [20,29]. We also observed an association of the AG Abs level with clinical manifestations and its changes in patients with chronic hepatitis C [34]. Overall, these observations may be a sign of an adaptive AG IgG antibody response in individual patients with digestive tract pathology. 

## 3. Prognostic Potential of AG Abs

The relationship between the preoperative level of AG Abs and its dynamics with the survival rate, tumour stage, and grading are noteworthy regarding prognostication. The combined determination of two independent anti-TF and -αGal Abs levels is more informative and can increase the prognostic potential (Figure 1E). The association with survival has been shown by the commonly used ELISA method with synthetic PGs. The glycan microarrays are a powerful tool for bioinformatics, enabling simultaneous detection of the Abs binding with a large set of glycans in a single format [4,35]. The diagnostic and prognostic potential of printed and suspension glycan arrays has been demonstrated for ovarian cancer, which may be a useful adjunct to the commonly used tumour marker CA125 [36,37]. Arrays enable the finding and selection of suitable glycans and also exploration of their optimal combinations to increase the predictive potential of AG Abs in cancer.

## 4. Study of the AG Abs Specificity

The heterogeneity and wide variety of natural glycans are challenges when studying the specificity of AG Abs. Synthetic glycoconjugates with known structure, conformation, and spatial orientation, and the density of glycotopes are a precision tool in the study of glycan-binding proteins; therefore, they could be adapted for different aims in glycobiology and medicine. PGs and adequate controls have demonstrated good reproducibility, low background, and better discrimination between comparable parameters in ELISA. Homogeneous PG models enable the detection of glycotope-specific Abs [38]; therefore, they have certain advantages over natural antigens containing different determinants. 

To study the AG Abs specificity, Abs were affinity isolated from sera of patients with cancer using glycans bound to carriers. In general, AG IgGs in the indirect and competitive ELISA showed either the reactivity restricted by the key glycan used for isolation or cross-reactivity to similar glycans, and no polyreactivity towards irrelevant antigens [39,40,41] (peculiarities are discussed in [42]). Anti-TF IgG populations were isolated from the sera of long-term gastrointestinal cancer survivors. Abs varied in the reactivity and cross-reactivity to synthetic TF-related conjugates. In the majority of samples, the Abs were more specific to TFβ (Galβ1-3GalNAcβ) than TFα, and the terminal Galβ residue was essential for antibody binding [39,41]. Affinity-isolated Abs were able to bind to natural antigens, such as asialo-glycophorin and GA1 glycosphingolipid (anti-TF populations), glycosphingolipids isolated from pig kidney or rabbit erythrocytes (anti-αGal IgG), and A_di_/A_tri_ cross-reactive IgGs agglutinated human A-blood group erythrocytes [39,40,41,42,43]. The binding of αGal-reactive Abs to glycosphingolipids depends on the avidity of Abs. For example, anti-αGal IgG with high avidity for B_di_-PG binds to rabbit erythrocyte glycosphingolipids, similar to monoclonal Abs Gal-13, while cross-reactive anti-A_di_ IgG with lower avidity for B_di_-PG does not bind [40,44]. The range of IC_50_ for PGs with isolated Abs was from 2 × 10^−8^ to 7 × 10^−6^ M. These results were obtained by affinity chromatography using a solution of 8 M urea for elution of Abs at lower temperatures (4–10 °C), with gradual removal of urea by dialysis, which may be preferable in order to restore the activity of AG Abs. 

Cancer-associated mucins that express TF, Tn, and Sialyl-Tn glycotopes are implicated in immune modulation and metastasis. Poorly glycosylated mucins at the tandem-repeat region can generate cancer-specific immunodominant epitopes, but immunogenicity is abrogated in a higher density of saccharides in MUC-glycopeptides [45]. Possibly, some cancerous sera contain AG auto-Abs that are reactive to truncated glycans of the MUC family or to the TF/Tn glycoforms of IgA1 that are found in sera and tumour samples from breast cancer patients [46,47]. However, mucin-type TAGs appear to not be a target for the studied AG Abs, which can be explained by the presence of cluster glycoforms in the context of the Thr/Ser-peptide core, while AG Abs recognise discrete glycotopes in PGs with a glycan density of 10–20%. Anti-TF IgGs did not bind the conjugate of 80%-TF-PG. AG Abs have been found to bind short fragments from larger glycans but do not recognise the same fragment in the context of the whole natural chain [45]. In the competitive ELISA, affinity-purified anti-Tn and -Sialyl-Tn IgGs demonstrated weak or no binding to mucins isolated from malignant breast tumours, ovine submaxillary mucin, and asialo-ovine submaxillary mucin, in comparison with Tn- and Sialyl-Tn-PGs [41]. The binding of anti-TF IgG to mucins from breast cancer specimens was observed in only 14% of specimens. The binding ability of purified anti-TF and anti-Tn Abs to breast cancer tissues was demonstrated by other authors using an immunohistochemical method [48]. The authors used natural antigens (asialofetuin and ovine submaxillary mucin) and a plasma pool of donors to isolate Abs, resulting in Abs that were able to specifically recognise carcinoma tissues. Apparently, such a method should lead to the isolation of broadly specific Abs populations. We used carriers with the synthetic homogeneous TFα-ligand and serum from individual patients to isolate anti-TF Abs. The resulting populations of Abs were heterogeneous and differed in cross-reactivity to glycosphingolipid-related glycans (TFβ, GA1, and Gb5 trisaccharide) and asialo-glycophorin [22,41]. 

Thus, Abs that are specific to TAGs that were presented in PGs exhibited low specificity in relation to natural tumour-associated and tumour-derived mucins. Anti-TF IgG Abs showed high reactivity to mucins isolated from breast cancer tissue but for rare specimens, which presumably was due to the admixture of bacterial glycoconjugates, since a diverse microbiota can colonise malignant breast tumours [49]. 

## 5. Paradigm of Microbial Origin of AG Abs Stimulus

Gut microbiota are involved in the formation of AG Abs both in the intestine and in the blood circulation system. The emergence of Abs against self-glycans, including Abs related to autoimmune disorders, is based on the molecular mimicry between bacterial and human glycoconjugates upon stimulation with bacterial antigens [50]. The immunodominant regions and molecular patterns of glycans for many human pathogens remain poorly studied. Mammalian and bacterial glycosyltransferases can synthesise the same glycan linkages; however, there are considerable differences between mammalian and bacterial glycans since the latter are often composed of repeating units and contain carbohydrates and other structures that are irrelevant to mammalian cells. TF and other glycotopes were found in the O antigens of Gram-negative bacteria [51]. The repeating units in the coaggregation receptor polysaccharides of oral streptococci contain a host-like disaccharide motif, either Galβ1-3GalNAc or GalNAcβ1-3Gal, as well as the immunodominant external GalNAcα saccharide [52,53]. In humans, Abs against TF and Tn are produced in response to bacterial antigens [54]. Human natural anti-Tn IgM Abs were able to bind a restricted number of Tn-terminated oligosaccharides better than the parent monosaccharide; however, Abs bound several bacterial polysaccharides that have no structural resemblance to Tn [55]. True disaccharide TFα has been identified in the capsular polysaccharide structure of *Bacteroides ovatus* strains, but, in general, human intestinal bacteria rarely express true TFα [56]. Mice immunised with *B. ovatus* in the absence of adjuvants developed specific anti-TFα IgM and IgG antibodies, which were able to bind TFα-carrying human cancer cells. In humans, dietary supplementation of *B. xylanisolvens* was able to increase the serum level of TFα-specific IgM Abs [57,58]. 

Microbial pathogens may cause changes in the glycosylation profile of mucins. Acute *Helicobacter pylori* infection can be accompanied by a dramatic but transient loss in mucin oligosaccharides that may promote bacterial colonisation and persistence [59]. TF glycan is expressed in surface membrane glycoconjugates of *H. pylori*. A better survival rate was observed in *H. pylori* seropositive vs. seronegative patients with gastric cancer, especially in those with a high level of anti-TF IgG [60]. According to a recent meta-analysis, *H. pylori* infection is an indicator of better prognosis in the European population of gastric cancer patients [61]. 

Bacterial glycosidases modify host glycans, thus exposing hidden areas, and thereby may provoke the production of auto-Abs. The TF antigen is usually cryptic, but it is exposed on erythrocytes and renal glomeruli because of the neuraminidase A desialylation in *Streptococcus pneumoniae* infections [62]. Adhesins of some pathogenic bacteria can bind TF and Tn glycans of the host cells [63,64], which also might be one of the reasons for AG Abs production. It is noteworthy that human circulating IgGs contain specificities of self-glycans that are receptors for viral and bacterial pathogens and/or exotoxins [65]. 

The αGal epitope (Galα1-3Galα1-4GlcNAc-R) is not expressed in human cells, but the anti-αGal Abs is naturally generated [66,67]. Human anti-αGal IgG was found to bind Gram-positive and Gram-negative pathogenic bacteria. Cross-reactivity to blood group-related glycans was shown for anti-αGal IgG and, in general, Abs contain multiple subsets with reactivities beyond terminal αGal hapten. These subsets in concert target a wide range of microbial polysaccharides and may contribute to human protective immunity against infections [40,43,68,69,70]. According to our observations, the increased level of anti-αGal Abs in patients with cancer is presumably associated with pathogen-mediated chronic inflammation [20]. 

Although a better survival rate for cancer patients with elevated anti-TF and -Tn IgG levels has been found, an explanation for these observations in terms of a protective immune response against TAGs-expressed cancer cells remains speculative. It is possible that some populations of anti-TAGs auto-Abs can impede tumour progression in individual patients, but to confirm this assumption, personal investigation using autologous samples is required. Solid tumours are typically low auto-immunogenic owing to immunosuppression and adaptation to host immunity. The presence of a certain microbiota in a tumour microenvironment can recruit and activate immune cells [71,72]. The existing relationship between AG Abs and the gastrointestinal microbiota community is worthy of attention. There is a rather indirect relationship between patient survival and the studied AG Abs, mediated by the presence of microorganisms that promote or inhibit cancer progression [73].

## 6. Influence of Microbiota on Cancer Progression

### 6.1. Beneficial or Normalising Effects of Bacteria

The human microbiota is comprised of numerous bacterial genera, which are present in different areas of the body, mainly in the large intestine. The normal functioning of intestinal microbiota maintains immune homeostasis while dysbiosis and impairment of the intestinal barrier are associated with the development of inflammatory disorders, which may lead to cancer [74]. Bacterial microbiota can either inhibit or stimulate carcinogenesis and tumour progression via different mechanisms [75,76,77]. Opportunistic infections may occur in immunocompromised cancer patients that need antibiotic treatment. Using probiotics including *Bifidobacterium* and *Lactobacillus* genera may normalise microbial composition, protect against pathogen-mediated inflammation, and inhibit carcinogenesis [74,78,79,80]. Perioperative administration of probiotics/synbiotics in gastrointestinal cancer patients may improve the quality of life while reducing postoperative complications, including infection incidence, as was shown in meta-analyses [81,82].

Various tumour-related factors predicting the survival rate after tumour resection have been explored, one of which is tumour-associated microbiota. Prolonged survival may depend on the immunogenicity of tumour neoantigens, which can be recognised by T cell clones that are cross-reactive to microbial peptide antigens [83]. In patients with pancreatic cancer, the tumour microbiome diversity affects survival. In long-term survivors, the unique intra-tumoural composition of microbiota was identified, which may contribute to a favourable microenvironment that is characterised by the recruitment and activation of CD8 T cells [71]. In CRC patients with a better clinical outcome, tumours were significantly enriched in certain bacterial genera and exhibited a high expression of genes encoding T cell markers and chemokines [72].

### 6.2. Harmful Effects of Bacteria

Commensal microorganisms are crucial for the maintenance of local and systemic immune homeostasis, and disbalance of the commensal ecosystem resulting in dysbiosis [74]. Different taxa of tumour-associated microbiota are implicated in cancer pathogenesis and their diagnostic and prognostic potential have been assessed [84]. Dysregulated immune response in the tumour microenvironment and anticancer therapy-mediated systemic immunosuppression may also contribute to the colonisation of pathogens in tumours. Anaerobic bacteria can colonise hypoxic areas of tumours, which can be used for bacteria-based cancer immunotherapy [85]. Opportunistic and pathogenic bacteria propagate in dysbiosis and potentiate tumour progression through chronic inflammation, modulation immune responses, and metabolic and signalling pathways [76,77]. Dysbiosis can increase intestinal permeability and potentiate the translocation of microbial products through the mucosal barrier to the liver, spleen, and bloodstream, thus promoting inflammation, fibrosis, and tumourigenesis [77,86,87].

Some opportunistic and pathogenic bacteria can colonise and predominate in tumour tissues of the gastrointestinal tract compared to adjacent non-tumour or normal tissues, thus boosting tumour development and spread [88]. It has been found that colonisation of enterotoxigenic and pathogenic bacteria is associated with advanced CRC [89]. Two species of oral bacteria, namely, *Fusobacterium nucleatum* and *Porphyromonas gingivalis*, are examined in this section as an example, owing to a growing interest in their role in orodigestive cancer pathogenesis and progression [90]. In general, bacteria promote proliferation and invasiveness of cancer cells, angiogenesis in the tumour microenvironment, and induce resistance to apoptosis through different signalling pathways [91]. These oral bacteria are closely related to the promotion of cancer development in regions other than the oral cavity. Although many studies have revealed an association, rather than a causal link, between microbial composition and oncogenesis, the impact of some bacteria on tumour progression deserves special attention. 

*F. nucleatum* belongs to an anaerobic Gram-negative bacterium. It is typically resident in the oropharynx, inhabits other organs, coaggregates with other bacteria species in the oral cavity, and is related to periodontal disease. *F. nucleatum* is present in colorectal carcinoma tissue and possesses immunosuppressive activity via the mechanism of T cell response inhibition [92]. The *F. nucleatum*-enriched tumours of CRC are characterised by increased tumour growth and invasion, and pro-tumoural immune responses [93,94]. Prevailing in the CRC tissues, *F. nucleatum* is stably maintained in primary tumours and distal metastases. Mouse xenografts of human primary colorectal adenocarcinomas were found to retain viable *Fusobacterium* and its associated microbiome through successive passages [95]. As was shown by qPCR, Fusobacterial genomic DNA was present in distal metastases, which suggests bacterial tropism to CRC tumours. This can be explained by the evidence for the integration of bacterial DNA into the human somatic genome via intermediate RNA, which was found more often in (a) tumours rather than normal samples, (b) RNA rather than DNA samples, and (c) the mitochondrial genome rather than the nuclear genome [96].

Interestingly, the Fap2 adhesin of *F. nucleatum* interacts with TF (the peanut agglutinin-reactive glycan) in CRC specimens. This binding leads to the bacterial enrichment of primary and metastatic TF-overexpressed adenocarcinomas [63]. In mice models, hematogenous *F. nucleatum* can colonise colorectal and mammary tumours by the mechanism of Fap2–TF interaction, suppress the accumulation of tumour-infiltrating T cells, as well as promote tumour growth and metastatic progression, which can be counteracted by antibiotic treatment. Bacterium-dependent evasion of tumour from immune response may occur by means of the interaction of Fap2 with TIGIT and CEACAM1 receptors, leading to inhibition of T and NK cell activities [63,97,98,99]. Outer membrane proteins Fap2 and RadD can induce cell death in human lymphocytes [100]. The immunosuppressive effect of *F. nucleatum* in the microenvironment of colorectal tumours may be due to the promotion of M2 polarisation of macrophages through a TLR4-dependent mechanism [101]. *F. nucleatum* can adhere to the intestinal epithelium through other surface proteins, inducing inflammation and the recruitment of inflammatory cells, and creating an environment that favours tumour growth [102]. 

The amount of *F. nucleatum* DNA in the CRC tissue is associated with a shorter survival time and may potentially serve as a prognostic biomarker [103,104]. The prognostic potential and relation of *F. nucleatum* to clinical characteristics of CRC patients was assessed using meta-analysis. Worse overall and cancer-specific survival, as well as tumour growth and distant metastases, were associated with an abundance of *F. nucleatum* in tumours [105]. *F. nucleatum*-positive tumours were detected more frequently compared to non-tumourous tissues in gastric cancer patients, and the presence of bacteria in the Lauren classification diffuse type of tumours was associated with significantly worse overall survival [106]. *Fusobacterium* species status was independently associated with a worse prognosis of patients with pancreatic cancer [107]. It is considered that unfavourable microbiota in the pancreas can influence the tumour microenvironment and induce immune suppression. It is assumed that alterations in the pancreatic microbiota could prolong the survival of patients with pancreatic ductal adenocarcinoma and improve the response to immunotherapy and chemotherapy [108,109]. *F. nucleatum* prevails in esophageal squamous cell carcinoma (ESCC), compared to adjacent non-tumour tissues, and the higher level of bacterial DNA is significantly associated with cancer-specific survival. High levels of intratumoural *F. nucleatum* are significant for predicting a poor survival rate in patients with ESCC and correlate with a poor response to chemotherapy [110,111]. *F. nucleatum subsp. polymorphum* is the most abundant species in oral squamous cell carcinoma (OSCC) tissues, and the numerical strength of oral *Fusobacteria* increases significantly with the progression of OSCC [112,113]. 

The serum levels of IgA and IgG Abs against *F. nucleatum* in CRC patients are significantly higher than in healthy controls and are associated with the amount of bacterial DNA in tissues. The diagnostic potential of Abs was demonstrated in CRC; however, an association with the CRC risk was not confirmed for antibody responses to the *F. nucleatum* proteins [114,115,116]. Circulating plasma IgG and salivary IgA Abs against *F. nucleatum* as beneficial biomarkers in pancreatic tumour lesions are worthy of attention [117]. 

The tumour infiltration with T lymphocytes is an independent informative prognostic factor, which has been confirmed in a systematic review and meta-analysis of patients with CRC, but the role of tumour-infiltrating B cells and plasma cells remains controversial [118,119]. The tumour microbiome can favourably shape immune responses by increasing the density of CD3+ and CD8+ T cells and the number of Granzyme B+ cells in patients with long-term survival [71]. In contrast, colonisation of harmful bacteria, such as *F. nucleatum*, suppresses the T cell response, and the amount of bacteria is inversely associated with the density of CD3+ T cells in tumours [92,120]. Thus, in the study of the antitumour response of T cells in the microenvironment, an analysis of the bacteria composition is required because the targets of the immune response in tumour tissues may be of microbial origin, and some bacteria can suppress the response or decrease the density of T cells.

*P. gingivalis* belongs to the phylum *Bacteroidetes* and is an anaerobic Gram-negative pathogenic bacterium. *P. gingivalis* is regarded as a keystone pathogen in chronic periodontitis, causing both dysbiosis and dysregulated immune response. *P. gingivalis* has either a local effect in its native territory, the oral cavity, or a systemic effect in distant organs. The most important virulence factors are lipopolysaccharide (LPS), fimbriae, gingipains, and outer membrane vesicles. Current knowledge suggests the involvement of *P. gingivalis* in the aetiology of orodigestive cancer, and it may be a periodontitis-irrespective microbial marker for the risk of orodigestive cancer death [121,122]. 

The invasion of *P. gingivalis* in host cells may occur before or after malignant transformation of cells. Notwithstanding, the pathogen is present in significantly higher amounts in gingival carcinoma specimens than in normal gingival tissue [121]. Like *Fusobacterium*, this pathogen can survive and persist within cancer cells, causing their increased proliferation and tumour growth in vivo [123,124]. Intracellular antigens of *P. gingivalis* have been detected in biopsies from OSCC [125]. *P. gingivalis* can adhere and intracellularly invade colon cancer cells, promoting their gingipain-mediated proliferation through the activation of the kinase signalling pathway (MAPK/ERK) [124]. This pathogen is capable of disordering the host immune response through endopeptidase gingipain-mediated cleavage of IgG Abs and evading the host adaptive immune system by inhibiting IL-2 accumulation [126,127]. *P. gingivalis* can inhibit phagocytosis of the OSCC cell line by macrophages and induce the polarisation of macrophages into an M2 tumour-associated phenotype [128]. There is evidence that peptidoglycan of *P. gingivalis* is responsible for upregulation of the immune-regulatory receptor PD-L1 in human oral carcinoma cells, and thus may support evasion of oral carcinomas from the immune response [129].

*P. gingivalis* can promote the formation of distant metastases and chemical resistance to anticancer agents through induced inflammation [121,130], and the localisation of this bacterium in tumour tissues has been shown to be related to a poor survival rate of patients with OSCC [131]. *P. gingivalis* is present mainly in cancerous tissues of patients with ESCC, as has been shown using different methods. Infection was significantly related to poor differentiation, severe lymph node metastasis, and stage of ESCC, and inversely related to the overall survival rate. These observations suggest that *P. gingivalis* could be an etiologic agent and potential prognostic indicator of ESCC [132].

IgG and IgA Abs against *P. gingivalis* in the serum were found to be potential diagnostic and prognostic biomarkers for ESCC. The high serum levels of these Abs were significantly associated with worse prognosis of ESCC patients [133]. The higher serum level of IgG against *P.gingivalis* has tended to be associated overall with increased orodigestive cancer mortality (P trend = 0.06) [122]. A large European prospective cohort study showed a two-fold higher risk of pancreatic cancer in individuals with a high level of plasma IgG Abs against *P. gingivalis*. On the contrary, a cluster with overall higher levels of Abs against commensal (non-pathogenic) oral bacteria showed a 45% lower risk of pancreatic cancer [134]. 

A comprehensive analysis of the tumour microbiome in numerous tumours and their adjacent normal tissues across seven cancer types found that each tumour type has a distinct microbiome composition. Intracellular-active bacteria are present in both cancer and immune cells (CD45+). Moreover, the DNA of various bacteria was detected in solid tumours that have no direct connection with the external environment, taking into account measures to reduce the effect of external DNA contaminants on samples [49].

Considering the ability of bacteria to preferentially colonise tumour tissues and persist in tumour cells, the question arises of how the presence of bacteria and their adhesins can affect the results of a comparative assessment of TAG expression by immunochemical methods using TAG-specific monoclonal Abs. The question should be addressed to the expression of TF and other glycans that can cross-react with bacterial glycans or serve as receptors for adhesins [52,53,63,98].

Thus, dysbiosis and the intratumoural colonisation of opportunistic and pathogenic bacteria affect tumour development through complex relationships within the triad: tumour microenvironment–resident bacteria–immune response. A significant part of the humoral immune response comprises Abs against glycoconjugates of bacteria inhabiting the oral cavity and gastrointestinal tract. Alterations in the composition of serum glycoconjugates occur in cancer, and are related to the inflammatory responses and extracellular matrix remodelling accompanying cancer development [12]. Colonisation of tumours by pathogens appears to contribute significantly to these alterations.

## 7. Prospects and Challenges of Integrative Glycome and Microbiome Research

In recent years, various glycomic databases and tools have been generated [135]. Databases are available on glycan-binding proteins and glycan-mediated host–pathogen interactions. The Carbohydrate Structure DataBase http://csdb.glycoscience.ru/database/ (accessed on 19 January 2021) contains the structures of glycans and glycoconjugates of bacteria, archaea, plants, and fungi. However, to date, integrated resources covering knowledge of the interaction of glycans and Abs have not been created. Microarrays based on synthetic carbohydrates that are inherent in human and microbial glycans for the determination of natural Abs and a humoral immune response against pathogens have been presented [4,35,136]. As was examined in Section 6.2, the determination of circulating IgG and IgA Abs against pathogens is worthy of attention as potential diagnostic and prognostic biomarkers for orodigestive cancer. The limited accuracy of diagnosis and prognosis in immunoassays may be due to the use of heterogeneous bacterial antigens and the cross-reactivity of immunodominant regions in different species. Some IgG-reactive glycan structures may be present in human pathogenic and nonpathogenic bacteria [137]. Using synthetic analogues and their combinations may improve the accuracy in the determination of Abs against pathogens, but, to date, the glycan structures of many human pathogens remain unexplored.

Innovations in high-throughput technology have prompted the development of computational and statistical analysis of accumulated bioinformatics to address the complexity of the adaptive immune repertoire and better understand the dynamics of adaptive immunity in different diseases, including cancer [138]. The microarrays with several hundred glycans may facilitate the selection of AG Abs related to tumour-colonising bacteria and contribute to a better understanding of their role in cancer pathogenesis. Further, 16S rRNA gene amplicon analyses and PCR are used for the identification of bacteria and the study of microbial composition and diversity. The culture-independent method of next-generation sequencing (NGS) has tremendously simplified 16S rRNA gene sequencing to providing comprehensive data [139]. Using NGS to determine tumour-associated microbiota, together with the study of AG Abs profiling and immune response, can elucidate the role of microbiota and AG Abs in cancer pathogenesis and provide clinically relevant bioinformatics [73].

## 8. Concluding Remarks

The high-throughput profiling assay for AG Abs, together with microbiota analysis using NGS techniques, can contribute to the development of new non-invasive biomarkers for diagnosis, prognosis, screening, and proper treatment of cancer. However, the identification of universal microbial markers is difficult due to the complexity and high variability of the microbiota composition between individuals [89]. The complex network of multifactorial interactions in cancer, the molecular heterogeneity of tumours, and the individual profile of AG Abs and the microbiome are challenges for population studies and can generate many speculations. Dynamic studies on the relationship between the humoral immune response and the microbiota composition in patients, stratified by tumour pathology, appear to be more relevant for further clinical use (Figure 2). The molecular heterogeneity of malignant tumours, including genomic instability, mutations, and profile of glycosylation, largely predetermines their further development and spread, as well as overcoming the body’s defence mechanisms and the response to therapy [12,140,141]. Commensal microorganisms maintain immune homeostasis and tolerance and can modulate the response to anticancer therapy [74]. Glycomic and microbiome research provide valuable information, but both are in need of unification and analysis that are oriented towards clinical alignment. The computational outputs based on the “multiomics” databases can provide unified bioinformation for monitoring and timely appropriate treatment, and open new opportunities for personalised oncology.

## Figures and Tables

**Figure 1 ijms-22-11608-f001:**
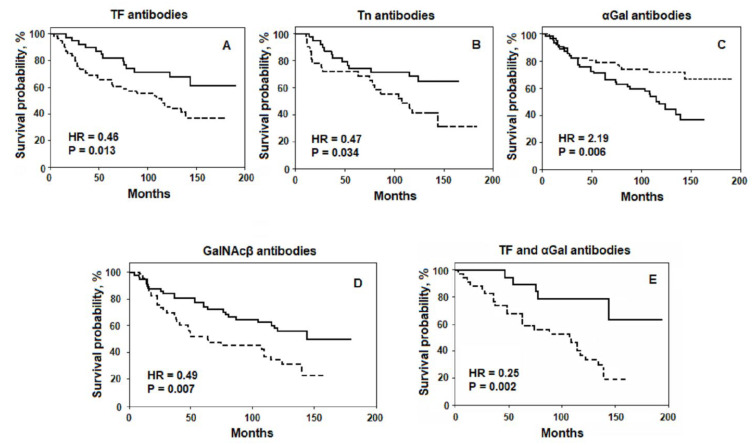
The relation of the preoperative AG Abs level to the survival rate of patients with gastrointestinal cancer. Solid line: level of serum AG Abs above or equal to the median; dashed line: the level below the median. (**A**,**C**) all patients; (**B**) patients with well and moderately differentiated G1-2 tumours; (**D**) patients with T2-4 size of the primary tumours. (**E**) the probability of survival assessed by two parameters: patients with increased anti-TF and decreased anti-αGal Abs level (solid line) vs. patients with decreased anti-TF and increased anti-αGal Abs level (dashed line). HR: hazard ratio. Adapted from [20,21]. (Under Creative Commons Act License).

**Figure 2 ijms-22-11608-f002:**
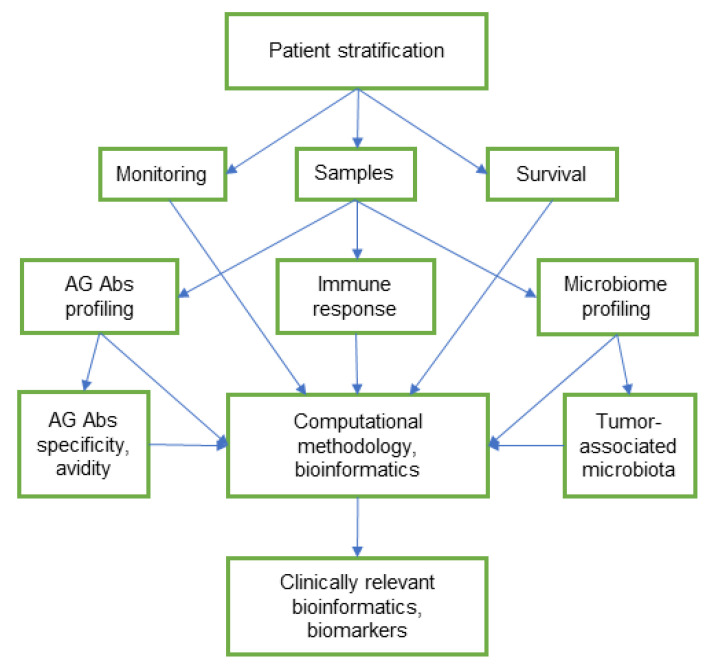
Conceptual scheme of integrative and clinically relevant studies of AG Abs and microbiome in cancer. Patient stratification by tumour localisation and histology, genomic mutations (tissue and liquid biopsies), and response to treatment (chemo/radiotherapy, immunotherapy, and targeted therapy). Monitoring by tumour markers (CEA, CA 19-9, CA 72-4, M2-PK), liquid biopsy, and commonly used clinical parameters. Survival: overall and cancer-specific survival. Samples: serum, plasma, and saliva for the AG Abs profiling; tumour tissues, saliva, and faeces for the microbiome profiling; tumour tissue and blood to assess the immune response. AG Abs profiling: using the glycan microarrays, differentiation of auto-AG Abs and Abs against microbiota. AG Abs specificity and avidity: affinity chromatography, surface plasmon resonance, modified and competitive ELISA. Microbiota profiling by 16S rRNA gene amplicon analyses and PCR. Tumour-associated microbiota: the recognition of beneficial species and pathogens. Immune response: the association with survival of the local immune response (tumour-infiltrating lymphocytes, macrophages, and neutrophils) and systemic response (a count and ratio of immune cells).

**Table 1 ijms-22-11608-t001:** Twofold or more changes of the AG Abs level in the monitoring of gastrointestinal cancer patients (presented for the first time).

AG IgGs	Total *n* cases	Increase ≥ 2, *n*	Decrease ≥ 2, *n*	Change in %
TF (Galβ1-3GalNAcα)	109	8	3	10
Tn (GalNAcα)	109	8	7	14
αGal (Galα1-3Galβ)	109	12	18	28
GalNAcβ	85	1	1	2
PF_di_ (GalNAcβ1-3GalNAcβ)	85	7	8	14

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
