# Peer review of "Prospects and Challenges of the Study of Anti-Glycan Antibodies and Microbiota for the Monitoring of Gastrointestinal Cancer"

_ijms, 2021, doi:10.3390/ijms222111608_

Round 1

Reviewer 1 Report

The authors describe the potential of anti-glycan antibodies as prognostic or diagnostic markers by providing examples of their utilization in several cancers. They continue to describe the pitfalls and issues with this approach. The review concludes with a description of developing an integrative approach that incorporates anti-glycan antibodies and tumor associated bacterial pathogens and their role on the tumor microenvironment.

The authors have made a good attempt at compiling currently available data. However, improvements in the overall organization is required to improve the value of the work.

  1. Line 50: (…many works on diagnostic and predictive studies of TAGs have been published) should include references of publications that could direct the reader to some of the literature available on this topic.
  2. A brief description of TF and Tn antibodies needs to be provided for readers who are not aware of glycan jargon.
  3. Line 50 and 51 seems contradictory. Using “However, the relationship between AG Abs and prognosis of cancer patients remains unexplored. For examples, patients with cerival…..)
  4. Better organization of 2. Association of the level of AG Abs with survival similar to section 6 could keep the reader engaged with the flow of study being described.

eg: start with some of the anti-glycan antibodies studied widely followed by their prognosis and the possible reasons for the prognosis.

  1. The concept of integrative glycome and microbiome analysis is an interesting idea. The authors have made a good effort at compiling the two ideas. However, further expansion and organization is required to keep the reader engaged.
  2. Furthermore, bacterial glycome could also be used as an approach to study this integration. It would be great to include studies that touch this aspect.

Author Response

Response to Reviewer 1 Comments

Dear Sir,

I am grateful to you for your review. In the revised version of the manuscript, I have taken into account your comments and remarks and made some changes in the manuscript.

Point-by-point answers:

Point 1: Line 50: (…many works on diagnostic and predictive studies of TAGs have been published) should include references of publications that could direct the reader to some of the literature available on this topic.

 Response 1: I have included references of publications (ref. [14-16]).

Point 2: A brief description of TF and Tn antibodies needs to be provided for readers who are not aware of glycan jargon.

Response 2: The author included a brief description of TF and Tn antibodies in the Introduction (lines 50-55). The interested reader can easily find more detailed information mentioned in references [5-9].

Point 3: Line 50 and 51 seems contradictory. Using “However, the relationship between AG Abs and prognosis of cancer patients remains unexplored. For examples, patients with cerival…..)

Response 3: Thank you for remark. The author changed the text (see lines 67-72).

Point 4: Better organization of 2. Association of the level of AG Abs with survival similar to section 6 could keep the reader engaged with the flow of study being described.

eg: start with some of the anti-glycan antibodies studied widely followed by their prognosis and the possible reasons for the prognosis.

Response 4: Among the published articles, there are works on the study of the diagnostic potential of natural AG Abs (the phrase and references are added in lines 66-67). The reader can find additional information in reviews that I have included (ref. [14-18]). The purposes of the present review were focused on prognosis and relation of AG Abs to survival including our monitoring of cancer patients, and on microbiota involved in cancer pathogenesis. The relation of AG Abs to prognostic potential, survival and clinical parameters of cancer patients remains almost unexplored (lines 67-72). I have no references to reorganize section 2 similar to section 6 according to the purposes.

Point 5: The concept of integrative glycome and microbiome analysis is an interesting idea. The authors have made a good effort at compiling the two ideas. However, further expansion and organization is required to keep the reader engaged.

Response 5: The presented integrative approach could be further extended for the reader's interest. However, this requires a lot of work. I received an e-mail from the editor asking me to revise the manuscript and reply to the reviewers on October 19, the day the second review was completed. As the sole author of this review, I will not have time to reorganize manuscript within 5 days that the editor gave me for my revision.

Point 6: Furthermore, bacterial glycome could also be used as an approach to study this integration. It would be great to include studies that touch this aspect.

Response 6: Bacterial glycome of commensals and pathogens, and corresponding antibodies in human, are still poorly understood. This is examined in section 5 and section 7 (lines 525-556).

Remark: Additional proofreading of English (British English) was done by a qualified native English-speaking proofreader.

Thank you again,

Kind regards,

Eugeniy Smorodin.

Reviewer 2 Report

The submitted manuscript is well written and well organized, it presents the topic formulated in the title in an accessible way which should be of interest to the readers of the International Journal of Molecular Sciences.

Minor comments

The present introduction is intended for readers dealing with the topic of anti-glycan antibodies. However, a paragraph introducing readers to the topic of glycans in the body and the production of antibodies against them (basic information) and their biological significance would be valuable to those unfamiliar with this topic. Some of this information can be found in the manuscript, but the paragraph with the information I mentioned above is missing.

Please explain in the body text the abbreviations at their first use (AG Abs, TAGs, PGs, TF, Tn, among others)

Table 1 – please indicate the source of data presented in the table.

Author Response

Response to Reviewer 2 Comments

Dear Sir,

I am grateful to you for your review. In the revised version of the manuscript, I have taken into account your comments and suggestions and made some changes in the manuscript.

Comments and Suggestions for Authors

The submitted manuscript is well written and well organized, it presents the topic formulated in the title in an accessible way which should be of interest to the readers of the International Journal of Molecular Sciences.

Point-by-point answers:

Minor comments

Point 1: The present introduction is intended for readers dealing with the topic of anti-glycan antibodies. However, a paragraph introducing readers to the topic of glycans in the body and the production of antibodies against them (basic information) and their biological significance would be valuable to those unfamiliar with this topic. Some of this information can be found in the manuscript, but the paragraph with the information I mentioned above is missing.

Response 1: Thank you for remark. I have inserted the paragraph in Introduction (see lines 27-34).

Point 2: Please explain in the body text the abbreviations at their first use (AG Abs, TAGs, PGs, TF, Tn, among others)

Response 2: I have inserted the explanations. Тn is a generally accepted abbreviation that is widely used even in the titles of articles without explanation. I have used as “TF precursor” (line 44).

Point 3: Table 1 – please indicate the source of data presented in the table.

Response 3: Table 1 is presented by me for the first time (this is inserted in the text, line 133).

Thank you again,

Kind regards,

Eugeniy Smorodin.